# Real-Time Deepfake Detection in the Real World

## Abstract

Recent improvements in generative AI made synthesizing fake images easy; as they can be used to cause harm, it is crucial to develop accurate techniques to identify them. This paper introduces "Locally Aware Deepfake Detection Algorithm" (*LaDeDa*), that accepts a single $9 \times 9$ image patch and outputs its deepfake score. The image deepfake score is the pooled score of its patches. With merely patch-level information, LaDeDa significantly improves over the state-of-the-art, achieving around 99% mAP on current benchmarks. Owing to the patch-level structure of LaDeDa, we hypothesize that the generation artifacts can be detected by a simple model. We therefore distill LaDeDa into Tiny-LaDeDa, a highly efficient model consisting of only 4 convolutional layers. Remarkably, Tiny-LaDeDa has $375\times$ fewer FLOPs and is $10{,}000\times$ more parameter-efficient than LaDeDa, allowing it to run efficiently on edge devices with a minor decrease in accuracy. These almost-perfect scores raise the question: is the task of deepfake detection close to being solved? Perhaps surprisingly, our investigation reveals that current training protocols prevent methods from generalizing to real-world deepfakes extracted from social media. To address this issue, we introduce *WildRF*, a new deepfake detection dataset curated from several popular social networks. Our method achieves the top performance of 93.7% mAP on WildRF, however the large gap from perfect accuracy shows that reliable real-world deepfake detection is still unsolved.

## 1 Introduction

Deepfake images are a leading source of disinformation with government and private agencies recognizing them as a grave threat to society (National Security Agency, 2023; World Economic Forum, 2024). Recent improvements in generative models, such as DALL-E (Ramesh et al., 2021), StableDiffusion (Rombach et al., 2022), and Midjourney (mid, 2022), significantly lowered the bar of creating fake images. Malicious parties are exploiting this technology to spread false information, damage reputations, and violate privacy online. Recent studies (Chai et al., 2020; Isola et al., 2017; Geirhos et al., 2018) showed that although deepfakes are semantically similar to real images, they have subtle, low-level artifacts that are easier to discriminate. This suggests that detection methods may gain from focusing on low-level image features.

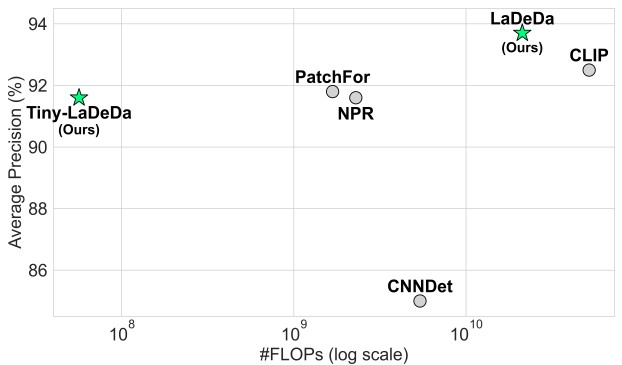

Figure 1: ***Performance vs. efficiency trade-off.*** Baselines comparison of average precision performance on real-world data as a function of floating-point operations per second (FLOPs) at inference time.

We therefore introduce *LaDeDa*, a patch-based classifier that leverages local image features to detect deepfakes effectively. LaDeDa's algorithm: i) splits an image into multiple patches ii) predicts a patch-level deepfake score iii) pools the scores of all image patches, resulting in the image-level deepfake score. To allow LaDeDa to work on small image patches, we use a variant of ResNet50 (He et al., 2016) that replaces some of the $3 \times 3$ convolutions by $1 \times 1$ convolutions. In particular, the best version of LaDeDa uses a receptive field of

$9 \times 9$, which we find is an effective size for deepfake [1] image detection. This encourages the classifier to focus on local artifacts rather than global semantics. Using only patch-level information, LaDeDa significantly improves over the state-of-the-art (SoTA), achieving around 99% mAP on the most popular benchmarks.

Since LaDeDa focuses on small patches, we postulate that a very simple model may be sufficient for detecting deepfake artifacts. To test this hypothesis, we design Tiny-LaDeDa, a highly efficient model consisting of only 4 convolutional layers. We train Tiny-LaDeDa by performing logit-based distillation (Hinton et al., 2015) using the patch-level deepfake scores predicted by LaDeDa (i.e., the teacher). Remarkably, Tiny-LaDeDa demonstrates superior computational efficiency compared to other SoTA methods (see Fig. 1), allowing efficient deepfake detection on edge devices with a minor decrease in accuracy.

With LaDeDa and Tiny-LaDeDa achieving an almost perfect score on current standard benchmarks, we ask whether deepfake detection is close to being solved. Arguably, the most popular source for spreading deepfakes is social media; we therefore test the performance of recent SoTA methods on deepfakes taken from social platforms. Perhaps surprisingly, we found that when using the current standard training protocols, SoTA methods (including ours) fail. Standard protocols attempt to simulate a real-world generalization, by training a detector using a single generative model (typically ProGAN (Karras et al., 2017)), and evaluate it across other generative models. However, the commonly used datasets in this simulation exhibit preprocessing discrepancies (e.g., real images are in lossy JPEG format while fake images simulated directly from a generator and saved in lossless PNG format), making the protocol less applicable for practical scenarios.

The failure in generalization to in-the-wild deepfakes persists even for methods that use post-processing augmentations that should make them robust to distribution shifts. To address these simulation imperfections, we introduce *WildRF*, a new deepfake detection dataset curated from popular social networks (Reddit, X (Twitter) and Facebook). WildRF serves as a comprehensive and realistic dataset that captures the diversity and complexities inherent online, which includes varying resolutions, formats, compressions, editing transformations, and generation techniques.

We validate the effectiveness of WildRF by retraining current SoTA methods on it. Our method achieves the top performance of 93.7% mAP on WildRF. Notably, it generalizes across social media platforms (e.g., training on Reddit images and evaluating on Facebook images) and is robust to JPEG artifacts, despite not using post-processing augmentations during training. The evaluation on WildRF shows that there is still a large gap from perfect real-world deepfake detection and highlights the importance of using a real-world benchmark.

To summarize, our main contributions are:

1. Introducing *LaDeDa*, a state-of-the-art patch-based deepfake detector for the real-world.

2. Distilling LaDeDa into Tiny-LaDeDa, a fast and compact, yet accurate student model for deepfake detection on edge devices.

3. Introducing the *WildRF* benchmark, extending deepfake evaluation to real-world settings, which are currently lacking in popular simulated datasets.

## 2 Related work

Deepfake detection aims to classify whether a given image was captured by a camera ("real") or generated via a generative model ("fake"). Two main paradigms exist to tackle this challenge.

**Deepfake detection by supervised learning.** These methods mostly focus on the architectural designs and discriminative features for discerning real and fake images. Wang et al. (2020) proposed using ResNet50 as a deepfake classifier, trained on real and fake images from one GAN method, and evaluate the performance on other GAN methods. PatchFor Chai et al. (2020) extends this idea, by learning a network that takes in a patch and outputs a deepfake score. While our method uses patches, similarly to PatchFor, we use knowledge

---

[1]In this paper, we use the term "deepfake" as in previous works, but mainly refer to AI-generated content.

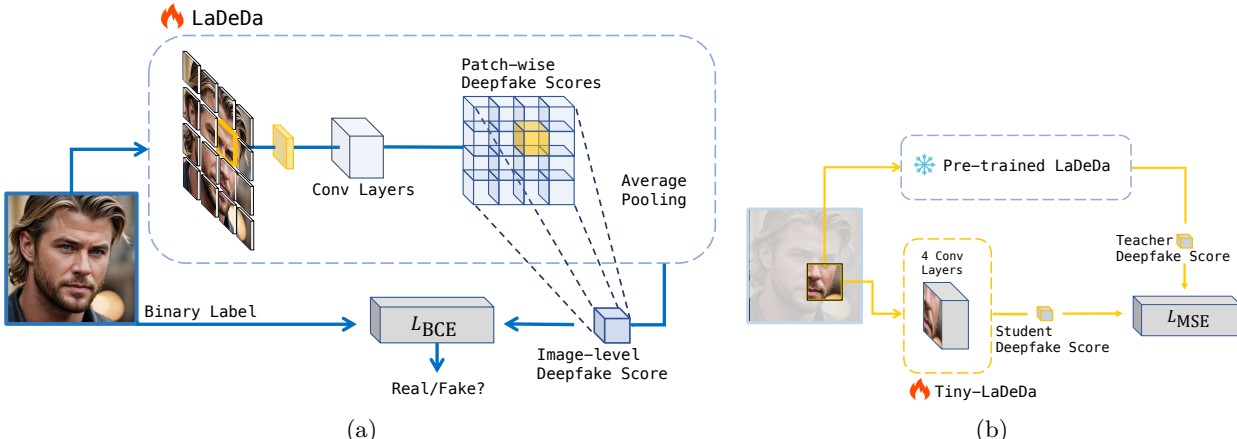

(a)                                      (b)

Figure 2: *(a) LaDeDa Training.* By limiting its receptive field to $q \times q$ pixels, LaDeDa yields a deepfake score for each $q \times q$ patch. The image-level deepfake score is the global pooling of the patches scores. We use binary cross entropy loss between the image label and its deepfake score. *(b) Tiny-LaDeDa Distillation.* Pre-trained LaDeDa (teacher) transfers patch-level deepfake score knowledge to train Tiny-LaDeDa (student).

distillation to estimate patch-level labels while PatchFor simply copies the image-level labels. Ojha et al. (2023) leverages the pre-trained feature space of CLIP (Radford et al., 2021), by performing linear probing on CLIP's image representations. Reiss et al. (2023) introduced the concept of "fact checking" for detecting deepfakes, while being training-free and relying solely on off-the-shelf features.

**Artifact-based detection methods** These methods leverage inductive biases in the image preprocessing stage to discriminate between real and fake images. Marra et al. (2019) revealed that GAN-based methods leave fingerprints in their generated images, which can be extracted using noise residuals from denoising filters. DIRE (Wang et al., 2023) focused on diffusion-based methods, measuring the error between an input image and its reconstruction, using a pre-trained diffusion model. Tan et al. (2023) introduced the NPR (Neighboring Pixel Relationships) image representation, aiming to capture the local interdependence among image pixels caused by the upsampling layers in CNN-based generators.

**Relation to PatchFor (Chai et al., 2020).** Both LaDeDa and PatchFor are patch-level methods that use deepfake detection datasets that only provide image-level labels. PatchFor labels each patch based on its image label with equal weights i.e., all patches from real images as equally real, and all patches from fake images as equally fake. Using a cross-entropy loss for each patch, PatchFor attempts to correctly classify all patches. However, not all patches are equally discriminative or needed for the overall image classification. Using an image BCE loss, LaDeDa has the extra flexibility of adaptive importance to different patches, putting emphasis on the patches that are more discriminative. Additionally, LaDeDa's patch scores can serve as soft labels for distillation (see Sec. **??** for the impact of our labeling strategy).

## 3 Method

**LaDeDa: Locally Aware Deepfake Detection Algorithm** Our core premise is that it is possible to discriminate between real and fake images with high accuracy based on low-level, localized features. This assumption is based on vast supporting literature (Chai et al., 2020; Isola et al., 2017; Geirhos et al., 2018; Frank et al., 2020; Mayer & Stamm, 2020; Zhong et al., 2023). We therefore introduce LaDeDa (denoted by $\phi$), a model that maps patches $p_i$ into a deepfake score $\phi(p_i)$, where higher values represent fake patches and lower values real ones. Average pooling the patch-wise scores of all image patches results in the image-level deepfake score:

$$S(I) = \frac{1}{N} \sum_{i=1}^{N} \phi(p_i) \tag{1}$$

where $I$ denotes the suspected image and $p_1, p_2..p_N$ its patches.

LaDeDa is a variant of ResNet50 He et al. (2016) (similar to BagNet Brendel & Bethge (2019)), but replaces most of the $3 \times 3$ convolutions by $1 \times 1$ ones, limiting the receptive field to $q \times q$ pixels (we use $9 \times 9$). The small receptive field forces LaDeDa to focus on local artifacts rather then global semantics, which we hypothesize is a good inductive bias for deepfake detection.

Specifically, for each image patch of size $q \times q$, LaDeDa infers a 2048-dimensional feature representation using multiple stacked ResNet blocks. The network applies a linear layer on the final patch-based representation, resulting in a per-patch score. The image-level score is the global pooling of the per-patch scores. Finally, we apply a sigmoid activation on top of the image-level score resulting in the predicted likelihood that the image is fake. We optimize the network parameters using the binary cross entropy loss:

$$\mathcal{L} = -\sum_{f \in \mathcal{F}} \log(\sigma(S(f))) - \sum_{r \in \mathcal{R}} \log(1 - \sigma(S(r))). \tag{2}$$

where $\mathcal{R}$ and $\mathcal{F}$ denote the real and fake image datasets respectively, $S$ denotes the network output (deepfake score) given an image, and $\sigma$ is the sigmoid function. See Fig. 2a for LaDeDa illustration.

While our default choice is to pool the patch-level scores using global average pooling, it is also possible (and sometimes desirable) to use other pooling operations. For instance, using max pooling effectively classifies the image based on the most fake patch. Using average pooling, results in classifying the image based on the collective characteristics of all patches. These patch-based deepfake scores can yield an interpretable way to visualize the patches that contribute the most for LaDeDa classification decision (see Sec. 6). Additionally, the linearity of the average pooling operation, makes the architecture distillation-friendly, as we will elaborate on in Sec. 3.

Unlike many previous approaches that rely on prior knowledge from large-scale datasets (e.g., ImageNet (Deng et al., 2009)), LaDeDa randomly initializes its parameters and does not require pre-training. For further details on the LaDeDa's architecture, see App. A.2.

**Tiny-LaDeDa**  Since LaDeDa operates on small patches, we hypothesize that a very simple model will suffice for detecting deepfakes . To this end, we propose Tiny-LaDeDa, a highly efficient model, obtained by distilling LaDeDa. As LaDeDa (i.e., the teacher) outputs patch-wise deekfake scores, we can train a simpler model (i.e., the student) to mimic the teacher's knowledge; aiming for similar performance, while being much more compact. Specifically, we leverage logit-based distillation (Hinton et al., 2015), which aims to transfer the knowledge encoded in the logit outputs (deepfake scores) of the teacher model to the student model.

To train Tiny-LaDeDa, we use a trained LaDeDa model to generate a distillation training set comprising of samples in the form of $(p_i, \phi(p_i))$ for each patch $p_i$ of each of LaDeDa's training images. We then train Tiny-LaDeDa to predict a patch-wise logit (deepfake score), using the MSE loss between the student's prediction and the teacher's output (see Fig. 2b). In logit-based distillation, the student is not limited by the teacher's architecture. Consequently, Tiny-LaDeDa uses only 4 convolutional layers with 8 channels each, yielding a model 4 orders of magnitude smaller than the teacher. Similarly to LaDeDa, at inference time, Tiny-LaDeDa outputs an image deepfake score, by pooling per-patch deepfake scores. See App. A.2 for further details on the Tiny-LaDeDa architecture. In Sec. 2 we elaborate on the differences between our method and PatchFor (Chai et al., 2020).

## 4   Is the task of deepfake detection close to being solved?

When training and evaluating LaDeDa using commonly used deepfake detection benchmarks (Ojha et al., 2023; Wang et al., 2020), it achieves a near perfect score of 98.9% mAP, significantly outperforming the current SoTA (Tab. 1). This naturally raises the question: is the task of deepfake detection virtually solved? Essentially, deepfake detection methods must be effective in real-world scenarios. As a sanity check, we evaluated them on a small sample of 50 images (25 real, 25 fake) taken from popular social networks (for a more comprehensive evaluation, see Sec. 5.1). Surprisingly, performance was near random, suggesting that

Table 2: ***Baseline performance - simulated protocol.*** All methods are trained on ForenSynth's train set (ProGAN), and evaluated on 16 generative models: (top) ForenSynth's test set (Wang et al., 2020), and (bottom) UFD's test set (Ojha et al., 2023).

| Method | ProGAN | | StyleGAN | | StyleGAN2 | | BigGAN | | CycleGAN | | StarGAN | | GauGAN | | Deepfakes | | Mean | |
|---|---|---|---|---|---|---|---|---|---|---|---|---|---|---|---|---|---|---|
| | ACC | AP | ACC | AP | ACC | AP | ACC | AP | ACC | AP | ACC | AP | ACC | AP | ACC | AP | ACC | AP |
| CNNDet (Wang et al., 2020) | 100 | 100 | 73.4 | 98.5 | 68.4 | 98.0 | 59.0 | 88.2 | 80.7 | 96.8 | 80.9 | 95.4 | 79.3 | 98.1 | 51.1 | 66.3 | 74.1 | 92.7 |
| PatchFor (Chai et al., 2020) | 99.6 | 99.9 | 93.9 | 99.2 | 94.5 | 99.6 | 74.4 | 89.6 | 85.3 | 93.5 | 76.3 | 90.4 | 64.7 | 92.4 | 89.2 | 92.8 | 84.7 | 94.6 |
| CLIP (Ojha et al., 2023) | 99.8 | 100 | 84.9 | 97.6 | 75.0 | 97.9 | 95.1 | 99.3 | 98.3 | 99.8 | 95.7 | 99.4 | 99.5 | 100 | 68.6 | 81.8 | 89.6 | 97.0 |
| NPR (Tan et al., 2023) | 99.8 | 100 | 96.3 | 99.8 | 97.3 | 100 | 87.5 | 94.5 | 95.0 | 99.5 | 99.7 | 100 | 86.6 | 88.8 | 77.42 | 86.2 | 92.5 | 96.1 |
| LaDeDa(Ours) | 100 | 100 | 100 | 100 | 100 | 100 | 90.1 | 96.5 | 98.9 | 99.8 | 93.7 | 99.7 | 91.0 | 99.2 | 68.1 | 95.6 | **92.7** | **98.9** |
| Tiny-LaDeDa(Ours) | 98.2 | 100 | 99.1 | 100 | 98.8 | 100 | 86.8 | 94.8 | 79.5 | 95.9 | 96.9 | 99.8 | 84.9 | 91.4 | 85.9 | 98.0 | 91.3 | 97.5 |

| Method | DALLE | | Glide_100_10 | | Glide_100_27 | | Glide_50_27 | | Guided | | LDM_100 | | LDM_200 | | LDM_200_cfg | | Mean | |
|---|---|---|---|---|---|---|---|---|---|---|---|---|---|---|---|---|---|---|
| | ACC | AP | ACC | AP | ACC | AP | ACC | AP | ACC | AP | ACC | AP | ACC | AP | ACC | AP | ACC | AP |
| CNNDet (Wang et al., 2020) | 52.5 | 66.8 | 54.2 | 73.7 | 53.3 | 72.5 | 55.6 | 77.7 | 52.3 | 68.4 | 51.3 | 66.6 | 51.1 | 66.5 | 51.4 | 67.3 | 52.3 | 68.4 |
| PatchFor (Chai et al., 2020) | 89.4 | 95.5 | 92.4 | 97.4 | 88.7 | 94.7 | 90.1 | 95.6 | 72.7 | 83.7 | 92.7 | 97.6 | 98.3 | 99.9 | 96.9 | 97.8 | 90.2 | 95.2 |
| CLIP (Ojha et al., 2023) | 87.5 | 97.7 | 78.0 | 95.5 | 78.6 | 95.8 | 79.2 | 96.0 | 70.0 | 88.3 | 95.2 | 99.3 | 94.5 | 99.4 | 74.2 | 93.2 | 82.2 | 95.7 |
| NPR (Tan et al., 2023) | 94.5 | 99.5 | 98.2 | 99.8 | 97.8 | 99.7 | 98.2 | 99.8 | 75.8 | 81.0 | 99.3 | 99.9 | 99.1 | 99.9 | 99.0 | 99.8 | 95.2 | 97.4 |
| LaDeDa(Ours) | 93.5 | 99.8 | 99.0 | 100 | 99.3 | 100 | 99.0 | 100 | 81.0 | 91.1 | 99.8 | 100 | 99.7 | 100 | 99.7 | 100 | **96.3** | **98.8** |
| Tiny-LaDeDa(Ours) | 80.2 | 98.4 | 98.5 | 99.9 | 98.8 | 99.9 | 98.9 | 100 | 76.2 | 87.8 | 99.4 | 100 | 99.3 | 100 | 99.1 | 99.9 | 93.8 | 98.2 |

the current evaluation protocol does not correlate with in-the-wild performance. We therefore reexamine the current evaluation protocols and suggest an alternative.

**Current: simulated deepfake detection protocol.** Most current methods follow a two-stage evaluation protocol. i) training a deepfake classifier using a set of "real" images and a set of "fake" images generated by a *single* generative model (typically ProGAN). ii) Evaluating the classifier on a set of real images and a set of generative models, most of which were not used for training. Many detection methods also use post-processing augmentations (e.g., JPEG compression, blur) during training to simulate unknown transformations an image may undergo before being encountered in-the-wild, potentially improving generalization. We refer to this as a *simulated* protocol.

Table 1: ***Baseline performance on current (simulated) protocol.*** mean average precision on 16 generative models from conventional benchmarks.

| Method | mAP |
|---|---|
| CNNDet (Wang et al., 2020) | 80.6 |
| PatchFor (Chai et al., 2020) | 94.9 |
| CLIP (Ojha et al., 2023) | 96.3 |
| NPR (Tan et al., 2023) | 96.7 |
| LaDeDa(Ours) | **98.9** |

**The simulated protocol is suboptimal.** Common datasets used in the simulated protocol comprise real images sourced from standard datasets (e.g., LSUN (Yu et al., 2015), ImageNet (Deng et al., 2009)) in JPEG format (lossy compression), whereas the fake images are generated and saved in PNG format (lossless compression). Training a classifier on such datasets can introduce bias towards differences in compression, leading to inflated perceptions of generalization performance when evaluated on test sets with similar biases (see Sec. 5.1). Moreover, simulating real-world artifacts through augmentations may fail to capture the full diversity of corruptions encountered in practice (see Sec. 5.1). In App. A.4.1 we provide further details on the commonly used datasets.

**WildRF: Aligning deepfake evaluation with the real-world.** We propose to improve deepfake evaluation and align it with the real-world by introducing *WildRF*, a realistic benchmark consisting of images sourced from popular social platforms. Specifically, we *manually* collected real images using keywords and hashtags associated with authentic, non-manipulated content (e.g., #photography, #nature, #nofilter, #streetphotography), and fake images using content related to AI-generated or manipulated visuals (e.g., #deepfake, #AIart, #midjourney, #stablediffusion, #dalle, #aigenerated). Our protocol is to train on one platform (e.g., Reddit) and test the detector on real and fake images from other unseen platforms (e.g., Twitter and Facebook). We denote this protocol as *social*. As both train and test data contain the type of variations seen in-the-wild, WildRF is a faithful proxy of real-world performance. See Fig. 3 for a WildRF overview.

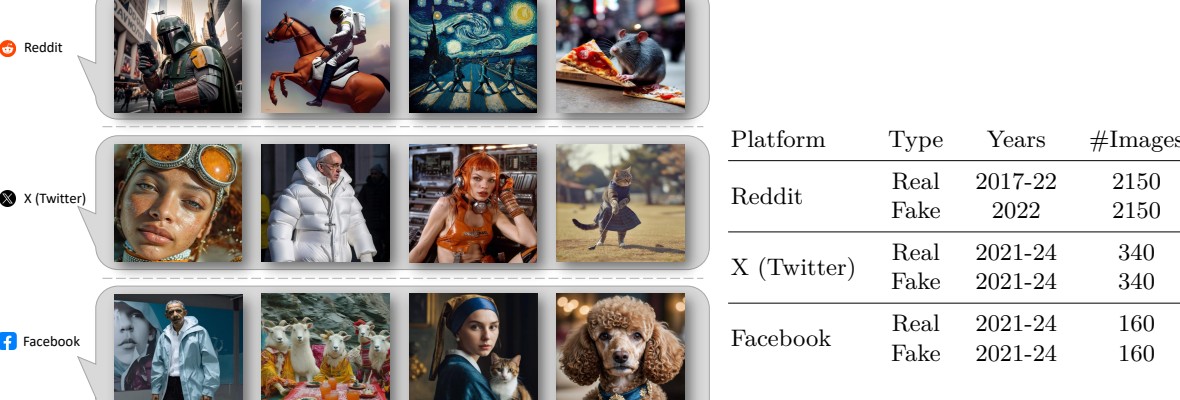

| Platform | Type | Years | #Images |
|---|---|---|---|
| Reddit | Real | 2017-22 | 2150 |
| | Fake | 2022 | 2150 |
| X (Twitter) | Real | 2021-24 | 340 |
| | Fake | 2021-24 | 340 |
| Facebook | Real | 2021-24 | 160 |
| | Fake | 2021-24 | 160 |

Figure 3: **_WildRF Overview._** A realistic benchmark consisting of images sourced from popular social platforms: _Reddit_, _X (Twitter)_ and _Facebook_. WildRF contains high variability in a range of attributes including image resolutions, formats, semantic content, and transformations encountered _in-the-wild_.

Table 3: **_Poor generalization to real-world data._** We show performance of SoTA methods trained on ForenSynth's train set (_ProGAN_) and evaluated on WildRF. It shows that training on the standard dataset, instead of in-the-wild deepfakes, generalizes poorly to in-the-wild images.

| Method | Reddit | | Twitter | | Facebook | | Mean | |
|---|---|---|---|---|---|---|---|---|
| | ACC | AP | ACC | AP | ACC | AP | ACC | AP |
| CNNDet (Wang et al., 2020) | 51.2 | 49.7 | 50.3 | 50.2 | 50.0 | 43.4 | 50.5 | 47.8 |
| PatchFor (Chai et al., 2020) | 63.9 | 74.0 | 47.8 | 51.3 | 68.2 | 75.3 | 60.0 | 66.9 |
| CLIP (Ojha et al., 2023) | 60.2 | 66.9 | 56.8 | 63.0 | 49.1 | 44.4 | 55.4 | 58.1 |
| NPR (Tan et al., 2023) | 65.1 | 69.4 | 51.7 | 52.5 | 77.8 | 86.3 | 64.8 | 69.4 |
| LaDeDa(Ours) | 74.7 | 81.8 | 59.9 | 67.8 | 70.3 | 90.1 | 68.3 | 79.9 |
| Tiny-LaDeDa(Ours) | 72.3 | 77.8 | 59.6 | 64.8 | 70.9 | 86.4 | 67.3 | 76.3 |

## 5 Experiments

We compare to SoTA baselines e.g., PatchFor (Chai et al., 2020), CNNDet (Wang et al., 2020), CLIP (Ojha et al., 2023) and NPR (Tan et al., 2023) using the standard metrics for evaluation: classification accuracy (ACC) (_threshold_ = 0.5), and average precision (AP).

### 5.1 LaDeDa performance under the current (simulated) protocol

We begin by comparing LaDeDa to the baselines using the current (simulated) protocol. For training, we use the standard train set of the ForenSynth dataset (Wang et al., 2020), which contains real images from LSUN (Yu et al., 2015), and fake images from ProGAN. For evaluation, we use 16 different generative models taken from the test sets of ForenSynth (Wang et al., 2020) and UFD (Ojha et al., 2023). The results in Tab. 2 demonstrate that LaDeDa and Tiny-LaDeDa outperformed the other baselines in terms of mAP (98.9%, 97.5% respectively), and LaDeDa also outperformed the baselines in terms of mean ACC (92.7%). For more details on the ForenSynth and UFD datasets, see App. A.4.1.

**Poor generalization to real-world data.** While methods trained on ProGAN achieve high performance on detecting deepfakes created by a large set of other image generators, they do not perform well when tested on real-world data. Specifically, we evaluate the baselines using our WildRF dataset (Sec. 4). The results in

Table 4: **JPEG compression bias.** Detectors train under the simulated protocol demonstrate lower performance when JPEG-compressing the test *fake* images. 100 JPEG quality = no compression, and 70 JPEG quality = lowest quality.

| Dataset | Quality | CNNDet (Wang et al., 2020) | | CLIP (Ojha et al., 2023) | | NPR (Tan et al., 2023) | | LaDeDa (Ours) | |
|---|---|---|---|---|---|---|---|---|---|
| | | ACC | AP | ACC | AP | ACC | AP | ACC | AP |
| StyleGAN | JPEG 100 | 83.3 | 96.2 | 88.0 | 98.5 | 97.6 | 99.8 | 100 | 100 |
| StyleGAN | JPEG 90 | 50.1 | 83.4 | 66.5 | 90.3 | 50.1 | 43.0 | 52.9 | 68.1 |
| StyleGAN | JPEG 80 | 50.0 | 71.3 | 56.8 | 84.6 | 49.9 | 38.0 | 50.3 | 54.5 |
| StyleGAN | JPEG 70 | 50.0 | 46.4 | 53.9 | 79.3 | 49.9 | 36.8 | 50.0 | 44.7 |

Tab. 3 show much lower performance than the test set of the simulated protocol. This gap highlights the limitations of the simulated protocol in estimating real-world performance.

**JPEG compression bias.** Many methods report their results on the simulated protocol (which is biased, see Sec. 4), with two detector variants: i) training with post-processing augmentations (e.g., JPEG compression, blur), and ii) training without these augmentations. We claim that the later variant, can be biased towards compression artifacts. To demonstrate this, we retrained the CNNDet and CLIP baselines on ForenSynth without JPEG and blur training augmentations. For NPR, we used its official released checkpoint that does not include augmentations. We then evaluated these detectors on 3000 StyleGAN images (1500 real, 1500 fake) from ForenSynth's test set, where we JPEG-compressed only the fake images. This setup allows us to examine whether compressing fake images impacts their classification as real ones. As shown in Tab. 4, the detectors' ability to correctly classify fake images decreases as a function of the compression rate, even at a relatively low compression quality of 90. Although the augmentation variants attempt to improve generalization by simulating the unknown transformations an image may undergo, in practice, the simulation is suboptimal (see Tab. 3).

## 5.2 Real-world deepfake detection

**LaDeDa performance under our (social) protocol.** We retrained and evaluate all methods using our proposed WildRF dataset. The train set comprises (1200 real, 1200 fake) images from Reddit, and the test set comprises (750 real, 750 fake) different images from Reddit, (340 real, 340 fake) images from X (Twitter), and (160 real, 160 fake) images from Facebook. The results in Tab. 5 show that training on real data is much more accurate than on simulated data.

**JPEG compression robustness.** Fig. 4a shows that LaDeDa is robust to a range of JPEG compression rates. Note that training our method on WildRF made it JPEG-robust without using post-processing augmentations during training.

**Blur perturbation robustness.** As per common protocol, we blur the images with a Gaussian filter of varying $\sigma$ values. Fig. 4b, shows that LaDeDa is generally robust to blur perturbations, even without training with such augmentations. Training LaDeDa with such augmentations, like other methods do, improve robustness further. Note that high noise values (e.g., $\sigma > 1$) result in a blurry image, making manipulation more noticeable than what an attacker would typically use.

## 5.3 Real-time deepfake detection

Some practical scenarios require deepfake detection to not only be accurate, but also computationally efficient for real-time inference at scale and on edge devices. Fast inference will only get more important as deepfake content continues to rapidly spread across online platforms. Here, we evaluate computational efficiency in terms of floating-point operations per second (FLOPs) and network latency (seconds). FLOPs are a

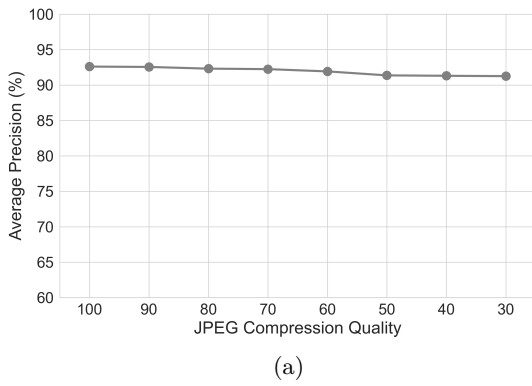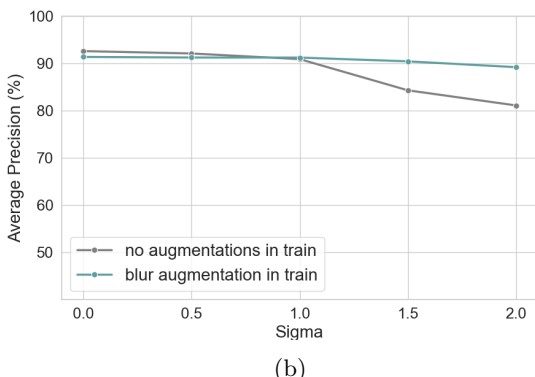

(a)                                        (b)

Figure 4: *(a) JPEG robustness.* We show LaDeDa average precision (AP) performance on facebook test set, as a function of JPEG compression quality from 100 (no compression) to 30 (high compression). *(b) Noise perturbation robustness.* LaDeDa shows robustness to blur perturbations, even without training with such augmentations. Training with Gaussian blur augmentations further improves robustness, even for images with $\sigma > 1$, which are more blurred than typical attacker manipulations.

Table 5: *Baseline performance - social protocol.* All methods are trained on WildRF train set (*Reddit*), and tested on WildRF test set. The results show a remarkable improvement compared to those achieved when trained using the simulated protocol.

| Method | Reddit | | Twitter | | Facebook | | Mean | |
|---|---|---|---|---|---|---|---|---|
| | ACC | AP | ACC | AP | ACC | AP | ACC | AP |
| CNNDet (Wang et al., 2020) | 75.4 | 86.8 | 71.4 | 84.1 | 70.6 | 83.5 | 72.5 | 85.0 |
| PatchFor (Chai et al., 2020) | 87.8 | 94.3 | 81.6 | 91.4 | 77.1 | 90.3 | 82.2 | 91.9 |
| CLIP (Ojha et al., 2023) | 80.8 | 94.2 | 78.1 | 93.1 | 78.4 | 90.6 | 79.1 | 92.5 |
| NPR (Tan et al., 2023) | 89.8 | 95.7 | 79.5 | 90.3 | 76.6 | 88.9 | 81.9 | 91.6 |
| LaDeDa(Ours) | 91.8 | 96.0 | 83.3 | 92.8 | 81.9 | 92.6 | **85.7** | **93.7** |
| Tiny-LaDeDa(Ours) | 84.5 | 92.4 | 82.3 | 91.7 | 80.7 | 90.4 | 82.5 | 91.6 |

hardware-independent measure of computational complexity, quantifying the total number of floating-point operations (addition, subtraction, multiplication or division) required for a single forward pass of a given model. Network latency measures the time it takes the model to process an input (e.g., an image) and output its prediction. Clearly, real-time deepfake detection needs low FLOPs and low latency. We simulate a low resource environment, similar to a mid-range smartphone with a single CPU core, and 4GB RAM. Tab. 6 shows that Tiny-LaDeDa with only a mild degradation in performance compared to LaDeDa, is $375\times$ faster and $10,000\times$ more parameter efficient.

## 6 Discussion and limitations

**Interpretability.** Since LaDeDa maps each $q \times q$ patch into a deepfake score, we can create a heatmap visualizing the most discriminative patches. In Fig. 5a, we show two such heatmaps of fake images, where areas with notable intensity changes tend to get high deepfake scores. Delving into the most fake patches (maximal deepfake scores), reveals that fake image patches appear smoother than those from real images (Fig. 5b). This aligns with studies (Durall et al., 2020; Corvi et al., 2023; Zhong et al., 2023) showing that generative models leave artifacts in high-frequency components due to the upsampling operation, making it

Table 6: ***Real-time deepfake detection.*** We show number of FLOPs, Parameters and Latency baselines comparison. In (red), we show the number relative to Tiny-LaDeDa, which demonstrates highly computational efficiency.

| Method | #FLOPs | #Parameters | Latency |
|---|---|---|---|
| CNNDet (Wang et al., 2020) | 5.40B(×95) | 23.51M(×18k) | 0.75 sec(×37.5) |
| PatchFor (Chai et al., 2020) | 1.68B(×30) | 0.191M(×150) | 0.21 sec(×10.5) |
| CLIP(Ojha et al., 2023) | 51.89B(×920) | 202.05M(×15k) | 9.37 sec(×470) |
| NPR (Tan et al., 2023) | 2.29B(×40) | 1.44M(×1.1k) | 0.35 sec(×17.5) |
| LaDeDa(Ours) | 21.23B(×375) | 13.64M(×10k) | 2.87 sec(×144) |
| Tiny-LaDeDa(Ours) | 0.0566B | 0.00129M | 0.02 sec |

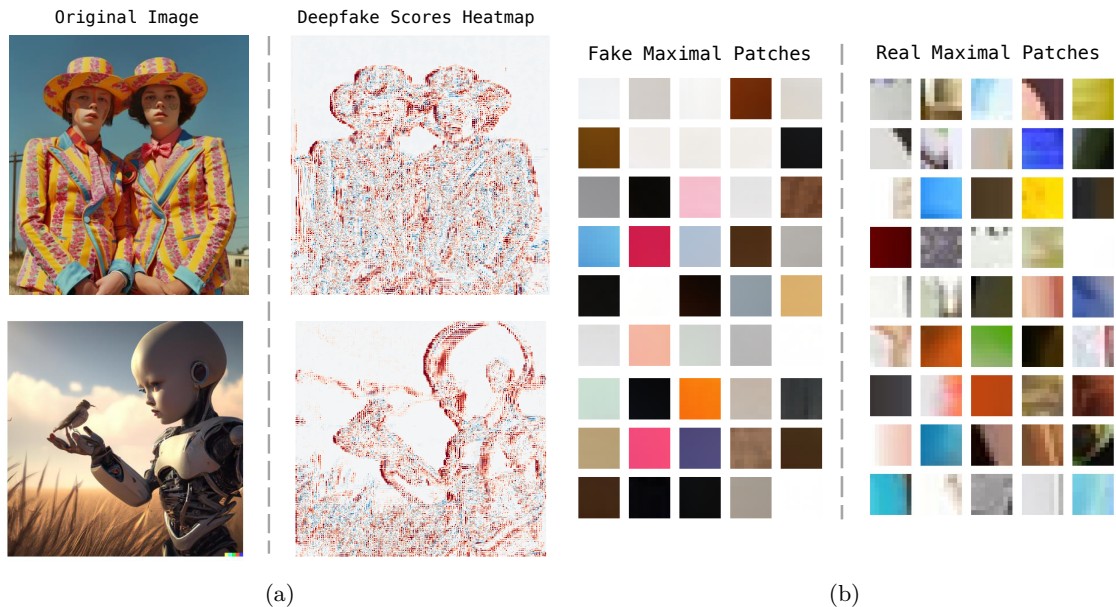

(a)  (b)

Figure 5: ***(a) Deepfake score visualization.*** High deepfake score in red, and low deepfake score in blue. ***(b) Maximal deepfake scores.*** We show the most fake patches (i.e., patches with highest deepfake score) in fake and real images. It appears that the most fake patches are smoother in fake images, compared to the most fake patches in real images.

difficult to synthesize realistic textures regions, thus smooth patches potentially smoother in fake images, and less smooth in real images, where a natural camera noise can appear.

**Size of WildRF.** WildRF's size is relatively small, with around 5000 images. Despite its size, WildRF serves as a valuable starting point for evaluating deepfake detection in real-world settings. Using it for evaluation has revealed that current simulated protocols hinder detectors from generalizing to deepfakes encountered on social media. While WildRF inevitably contains some biases, we expect these biases to reflect those encountered in-the-wild. To further examine WildRF's potential, we conducted a scaling law experiment, where we trained LaDeDa on subsets of increasing proportion (20%, 40%, 60%, 80% and 100%) of WildRF's training set and evaluated each instance performance on WildRF's test set. In SM we show that performance increases as a function of subset size. Importantly, the metrics have not saturated, indicating a room for improvement with larger dataset. While ideally, a larger and more comprehensive dataset would be beneficial, expanding WildRF is left for future work.

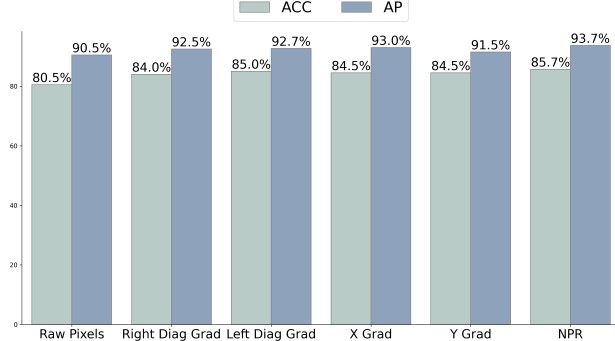

Figure 6: **Image preprocessing.** We show LaDeDa's performance using different image preprocessing as inductive bias.

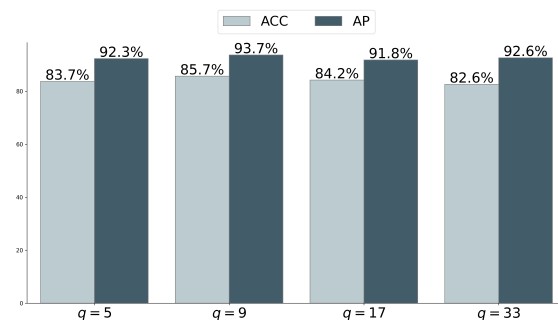

Figure 7: **Patch-size ablation.** We show LaDeD's performance using varying patch sizes.

Table 7: **Pooling operator ablation.** LaDeDa performance on the test sets of ForenSynth, UFD and WildRF, when using average pooling or max pooling to get image-level score from the patch-level scores.

| Method | Pooling | Test set | | | | | |
|---|---|---|---|---|---|---|---|
| | | ForenSynth (Wang et al., 2020) | | UFD (Ojha et al., 2023) | | WildRF | |
| | | ACC | AP | ACC | AP | ACC | AP |
| LaDeDa | Max | 90.1 | 99.1 | 88.9 | 97.0 | 84.9 | 92.4 |
| LaDeDa | Average | 92.7 | 98.9 | 96.3 | 98.8 | 85.7 | 93.7 |

**Generalization to near and far deepfakes.** When trained on a social network, our method and the baselines, generalize well to the other platforms. However, when trained on ProGAN/WildRF datasets the methods do not generalize well to the opposite dataset (WildRF/Simulated). To ensure that a single model can succeed on both protocols, we trained LaDeDa on a combination of 4000 ProGAN train images and WildRF train set, achieving comparable results to train and evaluate separately on each protocol.

# 7 Conclusion

We propose LaDeDa, a patch-based classifier that effectively detects deepfakes by leveraging local artifacts, and Tiny-LaDeDa, an efficient distilled version. Despite their high accuracy on current simulated benchmarks, we found that existing methods struggle to generalize to real-world deepfakes found on social media. To address this, we introduced *WildRF*, a new in-the-wild dataset curated from social networks, capturing practical challenges. While our method achieves top performance on WildRF, the considerable gap from perfect accuracy highlights that reliable real-world deepfake detection remains unsolved. We hope WildRF will drive future research into developing robust techniques against online disinformation that generalize to the real-world.

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

# A  Appendix

## A.1  Additional experiments

### A.1.1  Local is all you need?

While LaDeDa achieves SoTA performance by focusing on local artifacts, we ask if global features provide complementary information.

**Ensemble with CLIP (Ojha et al., 2023)**    We linearly ensemble LaDeDa with the CLIP baseline, which trains a linear classifier on top of the semantic CLIP features. The resulting score is:

$$S(I) = \text{LaDeDa}(I) + \alpha \times \text{CLIP}_s(I)$$

Where $I$ is an input image. The results in Fig. 8a show an increase of $\geq 3\%$ mAP on WildRF, when using the combined score, highlighting that semantic features are also useful for detecting deepfakes.

**Patches ensemble**    We trained 7 variants of LaDeDa with (5, 9, 17, 33, 65, 129, 257) patch sizes. To examine their importance, we set the weighted sum of the variants score as the image score. Equal weights (i.e., $\frac{1}{7}$ weight for each variant score) achieved 94.6% mAP on WildRF. Optimizing the weights on a validation set achieved 95.4% mAP, with smaller patches receiving higher weights. We further jointly trained 4 LaDeDa variants (9, 17, 129 and 257 patch size), as well as optimized their weighted sum, yielded a 96.1% mAP, with smaller patches again contribute more. Tab 8 shows that $9 \times 9$ patch-size LaDeDa achieves best AP of all patches, showing the effectiveness of small receptive fields. Still, there is benefit in using both high and low resolutions.

Table 8: ***Patches ensemble.*** Performance of LaDeDa with different patch sizes.

| Patch Size | 5 | 9 | 17 | 33 | 65 | 129 | 257 |
|:---:|:---:|:---:|:---:|:---:|:---:|:---:|:---:|
| **AP** | 92.3 | 93.7 | 91.8 | 92.6 | 91.2 | 90.3 | 88.9 |

## A.2   Architecture details

**LaDeDa architecture.**    The LaDeDa architecture is almost identical to the ResNet50 architecture, except for a few changes in the convolutional layers kernel sizes, strides parameters, and the final fully connected layer. We describe the architecture used for $9 \times 9$ patch-size receptive field. LaDeDa follows the standard ResNet50 design with four main residual blocks (layer1, layer2, layer3, layer4) consisting of bottleneck residual units. The first convolutional layer has a kernel size of 1x1 and 64 output channels, followed by a 3x3 convolution with the same number of channels. The residual blocks employ bottleneck units with 1x1 convolutions for dimensionality reduction and expansion. The downsampling operation is a $1 \times 1$ convolution with stride 2. The number of channels increases from 64 in layer1 to 256, 512, 1024, and 2048 in subsequent layers. Layer1 consists of 3 parallel bottleneck units, layer2 has 4 parallel units, layer3 contains 6 parallel units, and layer4 has 3 parallel units. Layer2 is the last layers that uses a kernel size of 3 (the layers after uses a kernel size of 1), thus limiting the receptive field of the topmost convolutional layer. After layer4, a global average pooling operation is applied, followed by a fully connected layer with a single output neuron and a sigmoid activation function, for binary classification.

**Tiny-LaDeDa architecture.**    The Tiny-LaDeDa architecture is a compact version of LaDeDa, designed for efficient performance with a reduced parameter count, using only 4 convolutional layers with 8 channels each. The architecture includes the following convolutional layers:

- $1 \times 1$ convolution (3 input channels, 8 output channels)

- $3 \times 3$ convolution (8 input channels, 8 output channels)

- $1 \times 1$ convolution (8 input channels, 8 output channels)

- $3 \times 3$ convolution (8 input channels, 8 output channels)

Tiny-LaDeDa results in a $5 \times 5$ receptive field. The final fully connected layer maps the 8 features to a single output, which is passed through a sigmoid activation function for binary classification. This streamlined architecture is lightweight and computationally efficient, making it suitable for real-time deepfake detection.

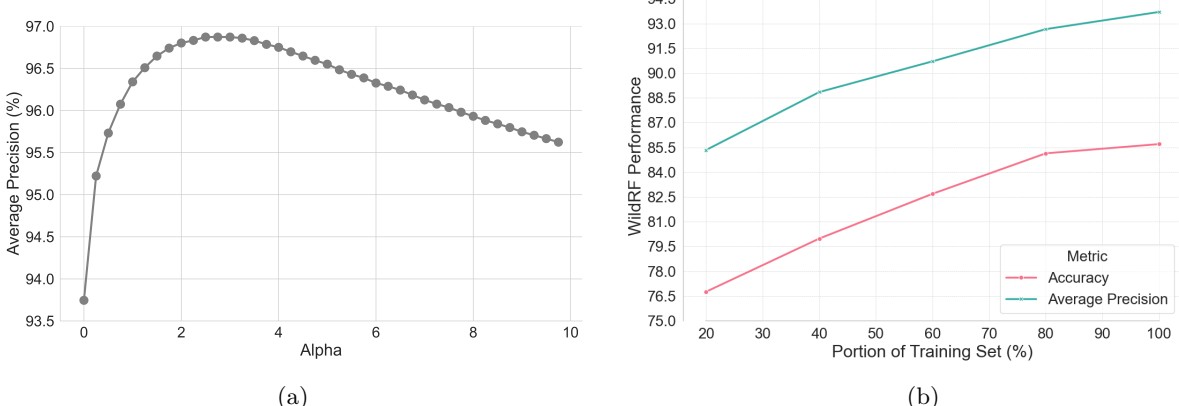

(a)                                                       (b)

Figure 8: *(a) Local and global deepfake scores.* We show average precision (AP) performance on WildRF, when ensemble LaDeDa deepfake scores with CLIP (Ojha et al., 2023) deepfake scores. *(b) Scaling law.* LaDeDa was trained on incrementally larger subsets of WildRF's training set and tested on WildRF's teset set. Performance increased with subset size, indicating room for improvement with a larger dataset.

### A.3   Implementation details

**Training LaDeDa.**   To train LaDeDa, we use the Adam optimizer (Kingma & Ba, 2014) with $\beta_1 = 0.9$, $\beta_2 = 0.999$, batch size 32 and initial learning rate of $2 \times 10^{-4}$. Learning rate is dropped by $10\times$ if after 5 epochs the validation accuracy does not increase by 0.1%, which is the same stopping criteria of (Wang et al., 2020; Ojha et al., 2023). During training, to have a uniform size, all images are resized to $256 \times 256$ resolution, and then randomly cropped to native size of $224 \times 224$ resolution. Note that we do not use post-processing augmentations (JPEG compression and Gaussian blur) as popular works (Wang et al., 2020; Ojha et al., 2023) do. During validation and test time, we directly resize the image to $256 \times 256$ resolution. As for the other baselines, we train them according to their official code repository. To train LaDeDa, we used a single NVIDIA RTX A5000 (21g).

**Training Tiny-LaDeDa.**   To train Tiny-LaDeDa, we use LaDeDa's patch-wise deepfake scores as soft labeling. We then use the Adam optimizer (Kingma & Ba, 2014) with $\beta_1 = 0.9$, $\beta_2 = 0.999$, batch size of 729 (which is the number of $9x9$ patches in an input image, and initial learning rate of $2 \times 10^{-4}$. To train Tiny-LaDeDa, we used a single NVIDIA RTX A5000 (21g).

### A.4   Simulated protocol

### A.4.1   Standard benchmarks in the simulated protocol

**ForenSynth (Wang et al., 2020) and UFD (Ojha et al., 2023) datasets.**   In this work, they chose ProGAN (Karras et al., 2017) as their single generative model in train set. Specifically, they used real images from LSUN (Yu et al., 2015), and fake images by train ProGAN on 20 different object categories of LSUN, and generate $18K$ fake images per category, resulting in $360K$ real and $360K$ fake images as the train set. As for the test set, they used ProGAN (Karras et al., 2017), BigGAN (Brock et al., 2018), StyleGAN (Karras et al., 2019), GauGAN (Park et al., 2019), CycleGAN (Zhu et al., 2017), StarGAN (Choi et al., 2018), Deepfakes (Rossler et al., 2019), SITD (Chen et al., 2018), SAN (Dai et al., 2019), IMLE (Li et al., 2019), and CRN (Chen & Koltun, 2017). Another work of (Ojha et al., 2023) has suggested the **UFD** dataset, comprising variations of diffusion models: guided (Dhariwal & Nichol, 2021), GLIDE (Nichol et al., 2021), LDM (Rombach et al., 2022), and DALL-E (Ramesh et al., 2021) as the fake images.

The real images sourced from the LAION (Schuhmann et al., 2021) and ImageNet (Deng et al., 2009) datasets. In this work they also utilize the train set of the ForenSynth dataset to train their detector. By examining the dataset publication of LSUN (lsu) (the real images in the ForenSynth train set), we can see that all images have been resized to $256 \times 256$ resolution, and JPEG compressed with quality of 75. Additionally, the real images in the UFD datasets are also in JPEG format. In 5.1 we show that methods that use the train set of ForenSynth dataset, can become biased towards JPEG compression artifacts, when tested on a test set with the same biases.

**GenImage Dataset Zhu et al. (2024)** In this dataset, the real images are all the images in ImageNet (Deng et al., 2009). The fake images was generated using 100 distinct labels of ImageNet. The train set fake images were generated using Stable Diffusion V1.4 (Rombach et al., 2022), and the test set fake images were generated using Stable Diffusion V1.4, V1.5 (Rombach et al., 2022), GLIDE (Nichol et al., 2021), VQDM (Gu et al., 2022), Wukong (wuk, 2022), BigGAN (Brock et al., 2018), ADM (Dhariwal & Nichol, 2021) and Midjourney (mid, 2022). In total, GenImage contains $1,331,167$ real and $1,350,000$ fake images. However, also here, we can observe the preprocessing discrepancy mentioned above. ImageNet images (the real images in GenImage) are in JPEG format, while the generated images in GenImage saved in PNG.

### A.4.2 Approaches for mitigating the datasets biases

A concurrent work of Grommelt et al. (2024) showed that training a detector on GenImage can cause it functions as a JPEG detector. To overcome this discrepancy, the authors suggested using an unbiased GenImage dataset where the real and fake images have similar resolutions and are JPEG compressed with the same quality factor. Chai et al. (2020) suggested to preprocess the images to make the real and fake dataset as similar as possible, in an effort to minimize the possibility of learning differences in preprocessing. To do so, they pass the real images through the data loading pipeline used to train the generator. As these approaches aim to mitigate the preprocessing differences between real and fake images, our approach uses images sampled from the distribution encountered in-the-wild, aiming to capture real-world artifacts differences between real and fake images.

### A.5 Related datasets extracted from social networking platforms

**Chen & Zou (2024).** In this work, they introduce a dataset of $800k$ ai-generated images with metadata from X (Twitter). However, the dataset is not opensourced and they did not provide real images, so we could not tested our method on it.

**Zi et al. (2020).** In this work, they introduce a dataset comprising 7300 face sequences, with more persons in each scene, and more facial expressions, compared to other deepfakes videos datasets. The faces extracted from 700 deepfake videos collected from video-sharing websites. As we were not able to get access to this dataset, we could not tested our method on it. You may include other additional sections here.

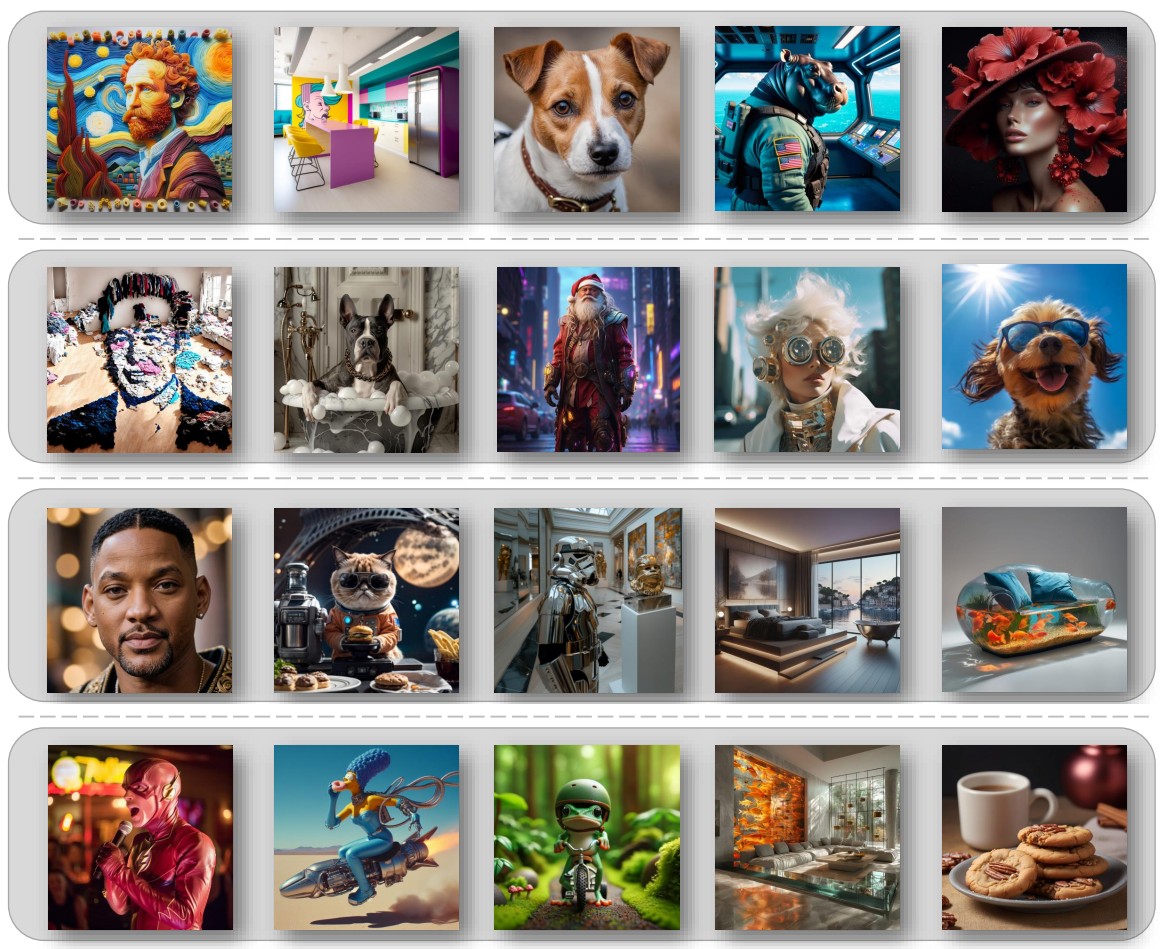

Figure 9: *Another WildRF image examples.*

