# OpenReview forum: "Real-Time Deepfake Detection in the Real World"
_TMLR — Withdrawn by Authors_

### Review · Reviewer_wqJZ · 2025-09-11

**Summary Of Contributions:**

The main contributions of the paper are 'LaDeDa', a patch-based deepfake detection method, along with its distilled version 'Tiny-LaDeDa', with 375x fewer FLOPs. The most significant contribution is the WildRF datasets for evaluating real-world deep fakes.

**Audience:**

Yes

**Audience Explanation:**

This paper demonstrates that current deepfake detection methods fail dramatically on real-world social media content despite achieving near-perfect performance on standard benchmarks. This exposes a critical gap in deepfakes detection literature.

The proposed dataset, WildRF, addresses this gap by providing images from actual social media platforms (Reddit, Twitter, Facebook) along with authentic compression artifacts and transformations.

**Broader Impact Concerns:**

The paper doesn't mention whether the data collection complied with the terms of service of Reddit, Twitter, and Facebook, which often have restrictions on automated data collection. Additionally, while the stated goal is detection, the proposed method can potentially be used to improve deepfake generation. Please provide a broader impact statement that covers these and other potential ethical concerns.

**Claims And Evidence:**

Yes

**Claims Explanation:**

- This paper makes a valuable contribution by highlighting the real-world performance gap in deepfake detection and providing a more practical dataset. It also proposes a detection method that generalizes well across previously proposed datasets in the literature.
- The patch-based design with 9×9 receptive fields is theoretically sound, focusing on local artifacts rather than global semantics.
- The computational efficiency of Tiny-LaDeDa is impressive.

**Requested Changes:**

Major
- There are many publicly available datasets for deepfake detection, yet the authors evaluate their method on only two. What motivated the choice of these specific datasets, and does the proposed detection method, 'LaDeDa', generalize to other deepfake detection challenges, such as celebrity faces, or identity-specific forgeries?
-  The paper points out that JPEG compression affects the results, but further investigation is required. The results show that accuracy drops sharply (to 50%) once the compression level reaches 90%. The link between compression quality and detection performance (Table 4) needs a more detailed study.

Minor
- Missing reference on page 3: "see Sec. ?? for the impact of our labeling strategy."
- Using hashtags like "#deepfake" and "#AIart" to identify fake content could introduce systematic biases and may not represent the actual distribution of deepfakes encountered in the wild.

---

### Review · Reviewer_FsQM · 2025-09-12

**Summary Of Contributions:**

This works introduces LaDeDa, a patch-based deepfake detector that constrains receptive fields using 1x1 convolutions and aggregates patch predictions via global average pooling, aiming to capture local artifacts in a simple, interpretable way.

---

### Strengths:
- The paper is well-motivated by the need for patch-level analysis in deepfake detection and the writing is very easy to follow.
- The inclusion of the new WildRF dataset is a valuable strength to this paper, as it better reflects in-the-wild distribution shifts by sourcing real and fake images from social platforms.
- The detailed explanation in the Appendix about JPEG compression artifacts and preprocessing discrepancies is much appreciated and very thorough.

---

### Weaknesses:
- The paper emphasizes in Section 2 that LaDeDa improves upon PatchFor by assigning adaptive importance to different patches, placing more weight on discriminative ones. However, in Section 3 (Equation 1), the image-level score is obtained by simple global average pooling of patch scores, followed by BCE loss at the image level. This pooling scheme enforces uniform weighting across patches, not adaptive weighting. Since 'adaptive weighting of patches' is framed as a core contribution distinguishing LaDeDa from PatchFor, this discrepancy represents a critical flaw.
- Based on the above weakness and given that the paper positions LaDeDa as a significant advance over PatchFor, the actual methodological novelty seems quite limited. LaDeDa seems to be, functionally, PatchFor trained with image-level loss instead of patch-level loss. Can the authors reiterate the major methodological differences between the two frameworks?
- In reference to the new dataset WildRF, did the authors check for identical or near-duplicate images reposted across platforms? Could such overlaps inflate the reported cross-platform performance?
- The authors emphasize how JPEG compression may have biased prior methods, particularly improving their performance on JPEG 70 (Table 4). However, even after retraining these SoTA baselines, their AP still surpasses LaDeDa, which additionally struggles at JPEG 90. Could the authors clarify why this occurs and why LaDeDa is more sensitive to compression?
- Could the authors elaborate on their decision to avoid transfer learning (i.e. reuse a pretrained ResNet on ImageNet) and explicitly forgo pretraining? What motivated this choice, and an ablation study to review this decision would be quite interesting.
- Especially since the paper questions if we have reached the limit of deepfake detection, it would be highly valuable for the community to see a formal (mathematical) definition of an upper bound on detection performance, and an assessment of how close LaDeDa or other SoTA methods are to this bound. While I suppose this would be challenging, the authors’ perspective on this would be greatly appreciated.

**Audience:**

Yes

**Audience Explanation:**

This work is definitely interesting to the wider audience of deepfake detection.

**Broader Impact Concerns:**

No broader impact statement is necessary. However, there is one ethical concern I would request the authors to address:

- Ethically speaking, the collection of real images for WildRF without explicit consent from the original creators (since this paper fails to mention this) is rather concerning. Could the authors clarify how issues of data ownership, copyright, and privacy were considered in dataset construction?

**Claims And Evidence:**

Yes

**Claims Explanation:**

The experiments and ablation studies provide solid support for the theoretical claims, with clear empirical evidence backing the main contributions. Nonetheless, there remain some concerns about the overall experimental validity of the work.

- The baselines used for comparison appear somewhat outdated. Please consider incorporating more recent methods, such as PUDD [1] and Loupe [2], for a fairer evaluation. You can also consider adding LNCLIP-DF [5], but it is a very recent paper.
- Beyond WildRF, there exist other datasets that capture social media scenarios, such as TrueFake [3] and SID-Set [4]. Could the authors clarify why WildRF was chosen over these, and consider reporting LaDeDa's performance on at least one of these datasets alongside 1 to 2 recent SoTA baselines?

---

[1] Pellcier et al. PUDD: Towards Robust Multi-modal Prototype-based Deepfake Detection. arXiv:2406.15921.

[2] Jiang et al. Loupe: A Generalizable and Adaptive Framework for Image Forgery Detection. arXiv:2506.16819.

[3] Dell'Anna et al. TrueFake: A Real World Case Dataset of Last Generation Fake Images also Shared on Social Networks. arXiv:2504.20658.

[4] Huang et al. SIDA: Social Media Image Deepfake Detection, Localization and Explanation with Large Multimodal Model. arXiv:2412.04292.

[5] Yermakov et al. Deepfake Detection that Generalizes Across Benchmarks. arXiv:2508.06248.

**Requested Changes:**

Critical changes are already discussed in the Weaknesses.

- There seems to be a missing reference in Section 2 in the paragraph "Relation to PatchFor".

---

### Review · Reviewer_cZLZ · 2025-09-13

**Summary Of Contributions:**

This paper introduces a patch-level deepfake detection method named LaDeDa, along with a distilled lightweight model, Tiny-LaDeDa. In addition, the authors propose a new dataset, WildRF, designed to better evaluate the performance of deepfake detection methods in the real-world scenarios.

**Additional Comments:**

NA

**Audience:**

Yes

**Audience Explanation:**

While some audiences working on deepfake would be interested in the paper, I find the the contribution of LaDeDa appears incremental and less attractive. Compared with PatchFor, LaDeDa mainly differs in the modification of labels during training, but does not introduce a fundamentally new idea. As a result, I believe audiences that works on deepfake yet have read PatchFor would only learn little from this paper.

**Claims And Evidence:**

No

**Claims Explanation:**

While Sections 4 and 5 present the experimental results and provide descriptions of the corresponding experimental setups, I think some of the paper's claims are not very convincing due to the following reasons:

- The paper does not provide a convincing rationale for why LaDeDa performs as it does. In particular, when compared with non–patch-level methods such as CLIP and NPR, it is unclear why their performance falls between that of PatchFor and that of LaDeDa. In other words, if patch-level approaches are indeed superior, then non–patch-level methods should not outperform PatchFor. If they are not, the authors should explain why LaDeDa still achieves an advantage.

- The paper should further elaborate on the differences between real-world and simulated evaluations. While the authors attribute the gap primarily to preprocessing discrepancies (e.g., image format differences), such issues appear relatively easy to address, for instance by standardizing the image format across both training and testing stages. A deeper discussion of more fundamental challenges underlying real-world scenarios would strengthen the motivation and contribution of the proposed dataset.

**Requested Changes:**

- The authors should further elaborate on the differences between real-world and simulated scenarios in deepfake detection.

- The authors should justify what new knowledge have be introduced to the community that some members would learn from the paper.

- The authors should further elaborate on the differences between real-world and simulated evaluations.

---

### Note · Authors · 2025-10-12

**Comment:**

We would like to express our gratitude to all the reviewers for their thorough and insightful feedback on our paper. However, after considering the overall scores we received, we have decided to withdraw the paper. We sincerely appreciate the time and effort the reviewers dedicated to evaluating our work.

**Withdrawal Confirmation:**

I have read and agree with the venue's withdrawal policy on behalf of myself and my co-authors.